# Recent Development to Explore the Use of Biodegradable Periodic Mesoporous Organosilica (BPMO) Nanomaterials for Cancer Therapy

**DOI:** 10.3390/pharmaceutics12090890

**Published:** 2020-09-18

**Authors:** Shanmugavel Chinnathambi, Fuyuhiko Tamanoi

**Affiliations:** 1Institute for Integrated Cell-Material Sciences, Institute for Advanced Study, Kyoto University, Kyoto 606-8501, Japan; chinnathambi.shanmugavel.8s@kyoto-u.ac.jp; 2Department of Microbio., Immunol. & Molec. Genet., University of California, Los Angeles, CA 90095, USA

**Keywords:** PMOs, biodegradable, drug delivery, cancer targeting and treatment

## Abstract

Porous nanomaterials can be used to load various anti-cancer drugs efficiently and deliver them to a particular location in the body with minimal toxicity. Biodegradable periodic mesoporous organosilica nanoparticles (BPMOs) have recently emerged as promising candidates for disease targeting and drug delivery. They have a large functional surface and well-defined pores with a biodegradable organic group framework. Multiple biodegradation methods have been explored, such as the use of redox, pH, enzymatic activity, and light. Various drug delivery systems using BPMO have been developed. This review describes recent advances in the biomedical application of BPMOs.

## 1. Introduction

Cancer is one of the most life-threatening diseases; it affects millions of people every year [1]. While the number of cases has decreased due to excellent diagnosis and treatment methods, the problem is that the current treatments kill healthy cells during chemotherapy, surgery, and radiation therapy. Targeting therapy is one of the best methods to reduce side effects. So, nanomaterials provide ideal ways to carry drugs to the targeted area without affecting healthy tissues. In the context of solid tumors that have a rich vasculature, the enhanced permeability and retention (EPR) effect can be exploited for cancer targeting [2,3]. In recent years, various nanomaterials were developed for biomedical solutions such as diagnosis and targeted drug delivery [4,5,6,7,8,9,10].

Mesoporous silica nanoparticles (MSNs) have emerged as a promising type of nanoparticles. They consist of an Si-O-Si framework with multiple pores generated by using surfactants such as cetyltrimethylammonium bromide (CTAB) as a templating agent. Because of the presence of the pore surface, these materials possess a large surface area where a variety of chemicals can be attached. In addition, their relative stability enables various chemical modifications to be carried out. Thus, sol–gel derived inorganic porous materials have become promising new carriers in drug delivery systems that have a high surface area, are well defined and tunable, and possess a uniform pore structure and biocompatibility [11,12,13,14,15,16].

Of particular interest are biodegradable periodic mesoporous organosilica nanoparticles (BPMOs) that were developed to enhance nanoparticle degradation in biological systems by incorporating a biodegradable organic bridge between two silicon atoms. BPMOs are a type of PMO, a unique class of materials that possess periodically ordered pores and nanometer-thick pore walls made of organosilica, in which each organic group is covalently bonded to two or more silicon atoms [17,18,19,20,21,22,23]. Because of their advanced structural features, PMOs are used for a broad range of applications [9,24,25,26,27,28]. In biomedical applications, fluorescently modified PMOs play critical roles in cell imaging and tumor targeting [25,29,30]. Several review papers have been published about the synthesis and applications of PMOs [16,17,18,19,21,22,24,25,27]. In this review, we focused on recent work that used BPMO systems for biodegradation studies, drug delivery, and anti-cancer treatment. 

## 2. PMO and BPMO

MSNs use tetraalkoxysilane (TEOS) as a precursor to form the Si-O-Si framework. On the other hand, PMOs [13,31,32] are prepared by using bridged alkoxysilane (Figure 1). In PMO, bi-silane has a functional organic bridge (R) between silicon atoms, like (R’O)3Si-R-Si(R’O)3, and OR`, representing a methoxy or ethoxy group (Figure 1). Those structures create remarkable advantages, such a as flexible framework, homogeneous distribution of organic units, and the convenient entrance of pore channels. Vercaemst et al. synthesized diastereoisomerically pure E-isomers and Z-isomers of 1,2-bis(triethoxysilyl)ethane for the development of PMO [33]. Asefa et al. synthesized PMO with an organic group inside the channel wall; PMOs contain bridge-bonded ethene groups directly integrated into the silica framework. This approach created a stable and periodic mesoporous ethene silica with high surface area and ethene groups that are readily accessible for chemical reactions [34]. Hoffmann et al. reviewed PMOs’ synthetic chemistry and its applications [22].

In the case of BPMOs, R in the precursor is used to design biodegradability into PMO. This is prepared by using a bridged alkoxysilane precursor that contains biodegradable chemical bonds such as ones that respond to redox conditions, low-pH conditions, enzyme conditions, etc. It is necessary to select an ideal degradable linker between two silicon atoms depending on the application of the nanomaterial. BPMOs are receiving significant attention in the development of nanomedical research and applications [10]. Biodegradation can be defined as the degradation caused by various conditions resulting from cell activities [16,35,36]. According to the US Food and Drug Administration (FDA), it appears that all injected agents need to be cleared from the body in a reasonable time period. A globular nanoparticle (NP) with a hydrodynamic diameter (HD) less than 6 nm can easily pass through the glomerular capillary wall, while it is difficult for those with a diameter more than 8 nm to cross through [37]. BPMOs can be designed to depend on internal conditions such as redox potential, pH conditions, presence of enzymes, temperature change, as well as on external stimuli such as light and ultrasound exposure.

It should be noted that R can also be used to enhance the loading and release of anti-cancer drugs. The loading of anti-cancer drugs can be significantly increased by various interactions created between the cargo and the nanoparticle. In addition, the release of anti-cancer drugs can be enhanced by a change in conditions, including low pH, redox conditions, and light exposure. By exploiting this feature, a variety of systems that respond to internal or external conditions have been developed.

## 3. Redox Responsive Systems

Redox cleavage of the sulfide bridges by an intracellular reducing agent such as glutathione (GSH) has been used for biodegradation. The idea to use redox conditions for biodegradation is based on the observation that the intracellular GSH level is much higher than that in the bloodstream (the intracellular GSH concentration can reach 10 mM, while the extracellular concentration is 10 μM). Thus, this type of BPMO is expected to remain intact during circulation in the bloodstream but will be degraded upon cellular uptake. Croissant et al. [13] reported biodegradable ethylene-bis(propyl) disulfide (EDIS)-based PMO nanorods and nanospheres for efficient in vitro drug delivery. Mixed PMO is controlled by the co-condensation ratio between bis(triethoxysilyl)ethylene and bis(3-triethoxysilylpropyl) disulfide. Michigan Cancer Foundation-7 (MCF-7) cell lines show perfect compatibility and biodegradability with physiological conditions when using EDIS NPs. The loading capacity of doxorubicin (DOX) in acidic conditions appears high in comparison with neutral conditions, and DOX is released in the lysosome upon stimulation with acidic stimuli. Additionally, the nanoparticles are localized after 24 h incubation. According to the above results, both materials acted as strong biodegradable Trojan horses for in vitro cancer therapy.

Vu et al. prepared BPMOs with tetrasulfide bonds and demonstrated their excellent capability to deliver doxorubicin. In this experiment, we used BPMO NPs carriers to deliver the anti-cancer drugs to a chicken chorioallantoic membrane (CAM) assay. Here, human ovarian carcinoma (OVCAR-8) cells on the CAM membrane of the fertilized egg were used to form the tumor. Later, DOX-loaded BPMO NPs were injected intravenously into the chicken egg to eliminate the tumor. No damage to the various organs present in the chicken embryo was observed. In opposition, widespread organ damage was observed when free DOX was used. The toxicity of the drug is dependent on the nanoparticle-associated delivery [38].

Recently, Mai et al. prepared BPMO nanoparticles containing tetrasulfide bonds by using two bridged alkoxysilane precursors, one of which included bridged tetrasulfide units. The BPMOs could be degraded completely after incubation with 10 mM GSH for three days, as demonstrated by microscopic observation as well as by dynamic light scattering (DLS) measurements (Figure 2) in both phosphate-buffered saline (PBS) and in simulated body fluid (SBF). Here, the authors used two models (tumor spheroids and the chicken egg tumor model) to evaluate the daunorubicin (DNR) as an anti-cancer drug. A very good accumulation of BPMO in the chicken egg model was observed, and this increased the uptake of the drug with a less toxic side effect. DNR BPMOs will be very good clinical candidates in the future [39].

Moghaddam et al. synthesized and characterized highly uniform redox-responsive polysulfide-based biodegradable silica nanoparticles that differ in size (58 to 332 nm), porosity, and composition (Figure 3a). These particles include mesoporous D that contains disulfide bonds, mesoporous T that contains tetrasulfide bonds as well as nonporous D that contains disulfide bonds. The release of DOX was analyzed in the presence and absence of GSH (Figure 3b). The degradation study demonstrated that the mesoporous nanoparticles underwent surface and bulk erosion and were degraded after incubation with 8 mM of GSH. Importantly, mesoporous nanoparticles had a higher degradation rate than nonporous nanoparticles, suggesting that the higher surface area of the mesoporous nanoparticles contributes to the degradation [40]. These nanoparticles did not exhibit significant cytotoxicity, as evaluated using murine macrophage cells (RAW264.7). Doxorubicin loading and release were examined.

Chen et al. reported biodegradable mesoporous organosilica nanosheets for the chemotherapy/mild thermotherapy of cancer with fast internalization, high cellular uptake, and high drug loading [41]. To compare these with silica-based materials, polyethylene glycol (PEG)-grafted copper monosulfide (CuS) @ mesoporous organosilica nanocapsules (MONs) are more efficient in the uptake of DOX (859 μg/mg). The ultra-thin porous nanosheet structure is responsible for the efficient degradation of PCMON, because 10 mM of GSH collapsed the PCMON. According to the above results, PCMONs are excellent biodegradable materials for drug delivery. In addition, with the above experiment, thermotherapy helps in the uptake and release DOX. In conclusion, PCMONs act like triple stimulus-responsive materials (pH value, GSH concentration, and laser irradiation); they accumulate in the cancer cells and destroy the tumor cells.

Wu et al. reported a hollow mesoporous organosilica nanocapsule (HMONs) framework containing disulfide bonds. PBS medium was used to confirm the disulfide bond breaks (biodegradation) in HMONs. Three days later, after dispersing the HMONs in GSH (10 mM), the authors used TEM to confirm the structural changes in the materials. Slight changes were noticed, but, seven days later, the HMONs’ shape and structure had completely collapsed. Two weeks later, the degradation was accelerated by the effect of reductive GSH-containing PBS. The hydrodynamic diameter of the materials showed decreased trends after 2 weeks. To support the above results, ICP analysis also showed a similar trend of Si presence. In conclusion, disulfide bonds containing HMONs have a high biodegradability [42].

## 4. pH or Enzyme Responsive Systems

Low pH conditions inside the endocytic pathway have been exploited for the degradation of nanomaterials and the release of the cargo, as discussed here. This approach is advantageous, as many tumor cells have lower pH values compared with normal cells. In addition, enzyme activities such as protease activity have been exploited for degradation.

Rao et al. prepared a cystamine-integrated periodic mesoporous organosilica (Cys-PMO). This nanoparticle possesses ordered hexagonal symmetry with a cylindrical shape and contains disulfide bonds. The loading and release of DOX were investigated, which revealed high DOX loading efficiency (50.6%) and the release of DOX at a low pH (Figure 4). Thus, this system is described as a dual (pH and redox)-sensitive system and appears to be promising as an anti-cancer drug for the intracellular cancer drug delivery system [43].

Zhang et al. synthesized PMOs that contained histidine amino acid (His-PMO) using pluronic P123 as a template. His-PMOs keep the ordered mesopores to uptake and release the drug compounds. Here, paclitaxel was used for anti-cancer drug delivery for an in vitro experiment. The high electrostatic interaction with the anti-cancer drug results in the high loading amount (28 mg/g). Finally, the results show that the release percentage and speed of paclitaxel (PTX) are completely dependent on pH values [44].

For controlled drug delivery applications, Motealleh et al. demonstrated PMO/alginate nanocomposite (NC) scaffolds in a pH-dependent manner. First, the anti-cancer drug was loaded into the pores and then the biopolymer coating was done. Later, the PMOs are embedded into an alginate network. The pH plays a major role in releasing the anti-cancer drug from the pores. In conclusion, the non-biopolymer-coated scaffold had a higher efficiency in delivering the drug to the cells in comparison with the biopolymer coating. However, the biopolymer coated scaffold was useful for slow and prolonged drug release [45]. Later, Motealleh et al. prepared self-assembled monolayers (SAMs) of pH-responsive chiral PMOs for drug delivery ability [46].

Moorthy et al. synthesized safranin–diurea-bridged hybrid mesoporous organosilica (SDU– HMS) for simultaneous diagnosis and therapy. The anti-cancer drug 5-fluorouracil (5-FU) was loaded by the interactions between the drug molecule and the NH-CO-NH part of the diurea ligands that exist in the pore walls. The drug release was low-pH dependent. The safranin fluorophore provided stable fluorescence in the nanoparticle. These experimental results proved that the SDU–HMS nanocarrier is desirable for red fluorescence-based in vitro tracking and pH-responsive anti-cancer drug release in cancer therapy [47].

Omar et al. used diazobenzene–triethoxysilyl amide precursors to synthesize BPMO nanoparticles that contained azobenzene linkers. Condensation was carried out with aromatic benzene or aliphatic ethane to produce an AZO-B (azobenzene linkers were condensed with aromatic benzene) or AZO-E (azobenzene linkers were condensed with aliphatic ethane) framework. This enabled the evaluation of the framework structure in relation to biodegradation, doxorubicin loading, and release. The authors showed that closer pore packing in AZO-B enhanced the enzymatic biodegradation of these hybrid frameworks compared with AZO-E. The above results show that a self-assembly pattern will be a useful drug delivery system in future [48]. The degradation of AZO-B and AZO-E can be catalyzed by azoreductase, which is known to cleave the azo-bond. This may have significance for colon-specific drug delivery.

Croissant et al. synthesized oxamide phenylene-based BPMO nanoparticles. The oxamide function provided biodegradability in the presence of trypsin that was used as a model enzyme. This system had excellent drug loading efficiency and was efficient in drug delivery to cancer cells [49].

Lu et al. reported yolk–shell structure-based PMOs for cancer cell targeting and drug delivery, which contained disulfide bonds. Near-infrared fluorescence (NIRF) dye was used on the surface of the materials with anti-Her2 affibody to target the cancer cells. Disulfide-containing PMOs are good candidates to deliver the anti-cancer drug DOX due to the GSH-dependent release. The above results indicate that PMOs are used for NIR Imaging, GSH, and pH-responsive drug release with cancer-targeting properties [50]. One year later, Lu et al. developed copper sulfide PMOs for hyperthermia-enhanced chemotherapy because of the photothermal effect of CuS. In addition, this material is effective for drug release in response to GSH, pH, and laser irradiation [51].

Moorthy et al. reported pyridine-modified PMOs for cancer therapy, which are called disilylated 2,6-dimethylpyridine-PMOs (DMPy-PMOs). They have enough efficiency to kill the cancer cells after 5-FU loading into the pores without toxicity. PMO materials have more efficiency in carrying drugs. They have the efficiency to kill cancer cells that depend on the pH and concentration of the materials. DMPy-PMOs were used due to their good absorbent properties in relation to removing metal ions [52].

Daurat et al. reported organosilica nanoparticles with either amine or ammonium walls constituting their structure. These nanoparticles are special in the sense that they can store and deliver gemcitabine or gemcitabine monophosphate (GMP), hydrophilic anti-cancer drugs that cannot be encapsulated in MSNs without using a pore cap. The release of GMP was dependent on a low pH. The nanoparticles were endocytosed by MCF-7 breast cancer cells and were efficient in causing cancer cell death [53].

## 5. Light and Ultrasound Responsive Systems

Shao et al. synthesized MoS_2_ nanosheet-covered PMOs (326 nm) with PEG decoration for NIR light-responsive drug delivery. The above materials uptake high-level DOX (160 µg·mg^−1^ PMOs) into the pores and have an excellent dispersion in physiological conditions. Upon light triggering, the materials have an excellent photothermal effect. Laser light of 808 nm breaks the MOS_2_ structure due to the photothermal effect, and drug release from the materials then occurs. The authors used two types of cancer cells (liver and breast) for combined photothermal therapy and chemotherapy. The final result showed more efficiency compared with chemotherapy or photothermal therapy alone [54].

Croissant et al. used PMO materials with an ethynylene bridge for photodynamic therapy and two-photon imaging applications. Here, a large two-photon absorber with four trimethylsilyl groups was used to capture NIR light. In addition, the PMO materials contained azidopropyltriethoxysilane in their structure. Later, the photosensitizer (propargylated fluorescent bromo-quinoline) was attached by click chemistry to generate singlet oxygen. After investigating the two-photon properties of the PMOs, the authors confirmed the fluorescence resonance energy transfer (FRET) from the fluorophore to the photosensitizer. The MCF-7 cell lines showed efficient cell death after irradiation with two-photon light [27,29].

Jimenez et al. prepared nanodiamond-encapsulated PMO, then anti-cancer drug doxorubicin was loaded in the pores. The drug release was dependent on pH. Later, two-photon exposure was used to generate singlet oxygen species. The above system worked in two ways to kill the cancer cells incubated with DOX-loaded ND@PMO [55].

Lin et al. developed protoporphyrin IX-encapsulated PMOs for light-induced metabolic inactivation of tumor cells. The synthesis involved the modification of the hydrophobic benzene group in the PMO framework by installing the hydroxyl group to incorporate photosensitizers. The above studies showed that the design of the post-modified nanocomposite was highly advantageous for in vitro photodynamic therapy (PDT) in human colon cancer cells (Figure 5) [56].

Wu et al. synthesized multifunctional PMO–DOX@MoS_2_-Polyethylenimine (PEI) – Bovine serum albumin (BSA)– Folic acid (FA) by wrapping MoS_2_ nanosheets onto DOX-loaded thioether-bridged PMO (PMO–SH) and then modifying them with the targeting moiety BSA–FA. The prepared PMO–DOX@MoS_2_-PEI–BSA–FA had a uniform diameter (196 nm) and high drug-loading capacity (185 mg·g^−1^ PMO–SH). Due to the excellent photothermal properties of MoS_2_ nanosheets, NIR laser irradiation resulted in the generation of heat, causing release of DOX in vitro and in vivo (Figure 6) [57].

Qian et al. reported hybrid hollow PMOs (HPMOs) for combined cancer therapy—ultrasound combined with chemotherapy. Because the hollow nanostructure enhances the ultrasound efficiency to change the acoustic microenvironment of the tissues, it received a large ultrasound energy deposition. The hollow sphere area was used as a reservoir for anti-cancer drug storage; then, drugs were released because of the ultrasound irradiation. The specific framework induced p–p supramolecular stacking between the benzene group-bridged framework and doxorubicin molecules. In addition, within the above experiment, the authors used HPMOs for cancer therapy in a mouse xenograft model. This system also shows good results in developing novel cancer therapy.

In Table 1, we summarize the PMOs and organo bridges discussed in this review for drug delivery applications. In addition, we also list the anti-cancer drug release systems using internal or external stimuli.

## 6. Related Studies

Thiol organosilica NPs represent a type of silica nanoparticles that are synthesized by the sol–gel method using a thiol organosilicate as a precursor. These nonporous silica nanoparticles are synthesized by using a precursor, 3-mercaptopropyltrimethoxysilane (MPMS). Doura et al. demonstrated that the use of a different precursor, 3-mercaptopropyl(dimethoxy)methylsilane (MPDMS), results in the preparation of nanoparticles that contain abundant disulfide bonds, as revealed by nuclear magnetic resonance (NMR) measurements and Raman spectroscopy [59]. They also used the combination of MPMS and MPDMS, which revealed that the ratio of disulfide bonds to thiols increased with the increase in MPDMS. The formation of disulfide bonds in the nanoparticles synthesized using MPDMS is due to their chemical composition, which enables the maximum number of two-siloxane bonds to be formed. Therefore, nanoparticles are formed only when disulfide bonds are formed within the framework. The degradation of nanoparticles containing disulfide bonds after incubation with 10 mM GSH for 7 days was confirmed using TEM analyses. A high ratio of MPDMS gives more degradation compared with free MPMS NPs.

Later, Mekaru et al. evaluated the biodegradability of disulfide-organosilica nanoparticles by soft x-ray photoelectron spectroscopy for cancer therapy implications [60]. They used two kinds of organosilica nanoparticles, MPMS and MPDMS, that were fabricated similarly to those described by Doura et al. [59]. Disulfide bonds are easily degraded by incubating with 10 mM GSH concentration at 37 °C for 7 days. The SEM analysis of MPDMS NPs showed biodegradation after 7 days, while MPMS NPs were not degraded. The soft X-ray photoelectron spectroscopy results of MPDMS before and after GSH treatment were consistent with the degradation of disulfide bonds by GSH. These results support the knowledge that MPDMS NPs possess a biodegradable feature that is advantageous for clinical translation in nanomedicine.

## 7. Summary and Future Prospects

PMOs contain organic–inorganic hybrid materials with highly ordered structures, uniform pore sizes, and a homogenous distribution of organic bridges in a silica framework. The addition of an organic bridge disrupts the periodic ordering of mesopores and monodisperse PMOs and expands the application of silica-based nanomaterials so that broad applications are possible.

Of particular importance are BPMOs that contain biodegradable bonds, including redox-sensitive bonds, pH-sensitive bonds, and bonds that are sensitive to enzymatic activities. A variety of studies have been carried out with BPMOs containing redox-sensitive bonds such as disulfide bonds and tetrasulfide bonds. The degradation of these nanoparticles under redox conditions was much faster compared to mesoporous silica nanoparticles without redox-sensitive bonds. In addition, the degradation of nonporous nanoparticles was slower, pointing to the importance of the mesoporous structure. Studies that demonstrate the in vivo degradation of BPMOs are limited, and this issue needs to be extensively investigated in the future, including excretion studies of BPMOs from an animal body. While this manuscript was being reviewed, a review paper discussing trends in degradable mesoporous organosilica-based nanomaterials for controlling drug delivery was published [61]. That review contains even more discussion on nonporous materials.

BPMOs have been shown to be effective in loading anti-cancer drugs and accomplishing the controlled release of these drugs. In this sense, it is important to point out that the presence of an organic moiety in BPMOs contributes to the loading of cargos to the nanoparticle due to the occurrence of additional chemical interactions between the cargo and the delivery vehicle. Furthermore, the release of the drugs can be carried out in a controlled manner, such as under reducing conditions, low-pH conditions, etc. Light-responsive release can also be carried out. This realization led to the development of various stimuli-responsive systems based on BPMOs.

## Figures and Tables

**Figure 1 pharmaceutics-12-00890-f001:**
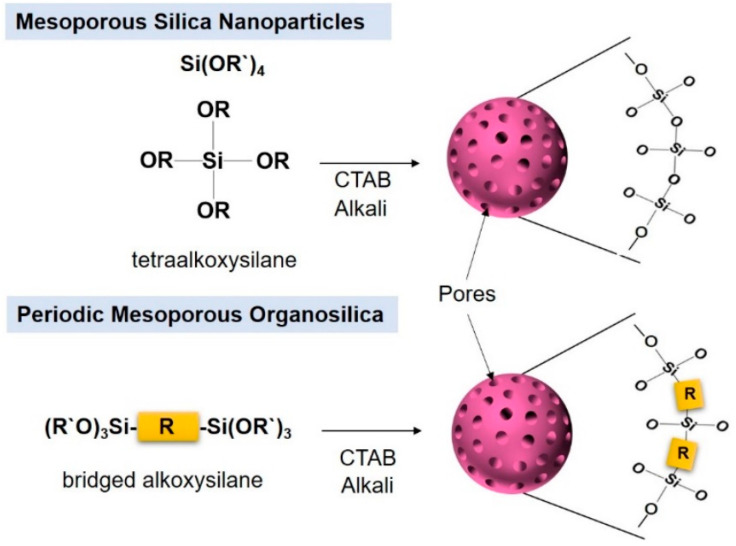
Schematic representation of mesoporous silica nanoparticles (MSNs) and periodic mesoporous organosilica nanoparticles (PMOs) with precursors.

**Figure 2 pharmaceutics-12-00890-f002:**
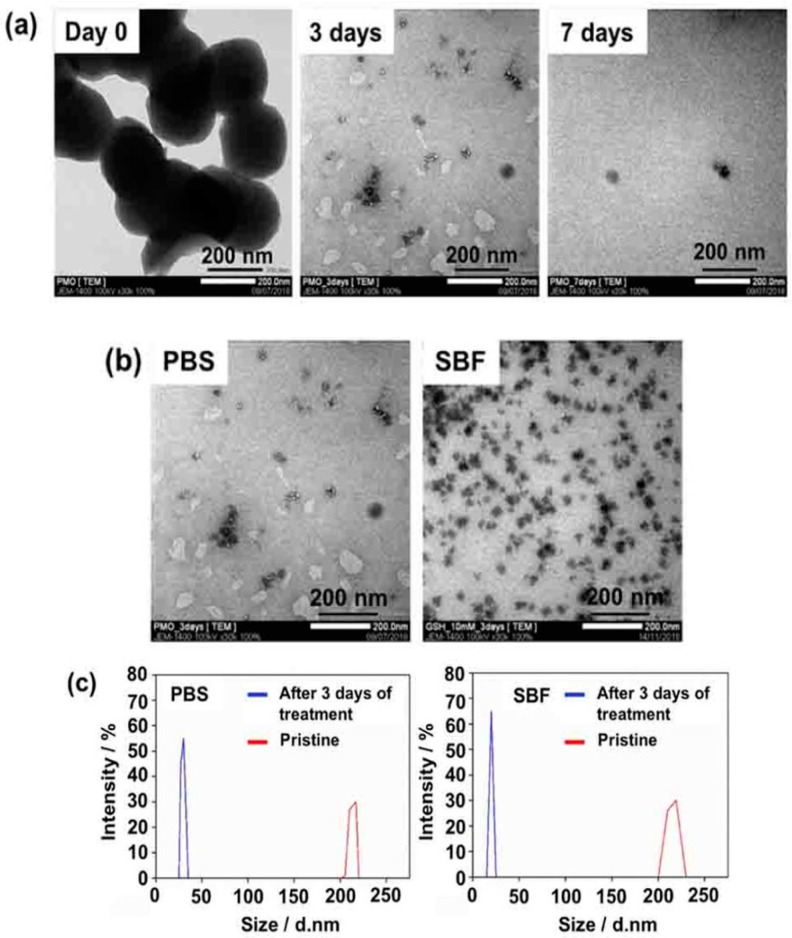
In vitro degradation of BPMOs: (**a**) TEM of degraded BPMO after incubating in PBS with glutathione (GSH) (10 mM) for various times. (**b**) TEM of BPMO after incubating with GSH (10 mM) in PBS or simulated body fluid (SBF) for 3 days. (**c**) Dynamic light scattering (DLS) measurements confirmed the average size of pristine BPMOs and degraded fragments after 3 days of treatment. Adapted from [39], Wiley Publications, 2020.

**Figure 3 pharmaceutics-12-00890-f003:**
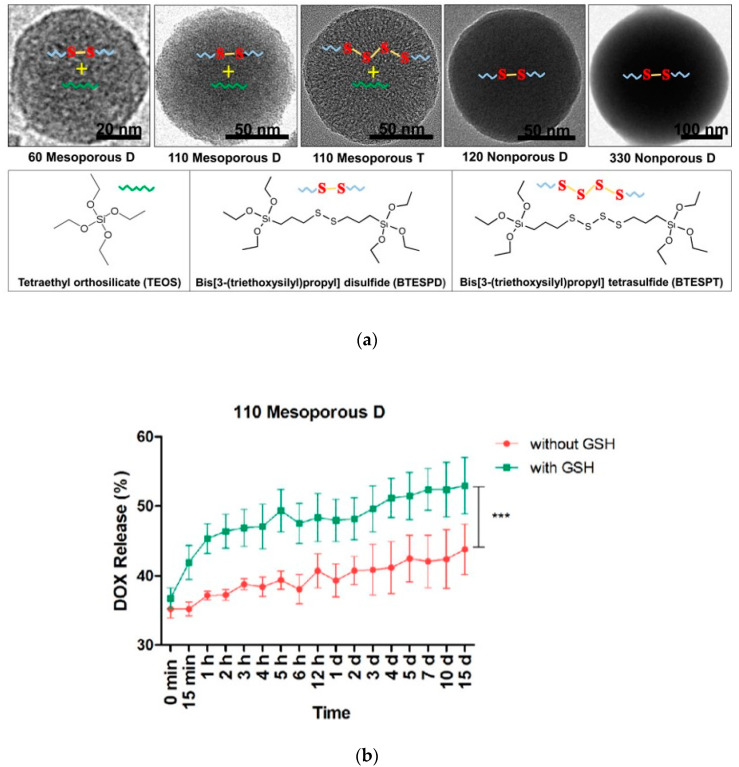
(**a**). Examples of degradable nanoparticles are prepared by Moghaddam et al. [40]. The top panel shows a transmission electron microscope (TEM) images. The bottom panel shows precursors used to prepare nanoparticles. Note that mesoporous D (disulfide-based SiO_2_ NPs) were synthesized by using the combination of two precursors, tetraalkoxysilane (TEOS) and bis-(triethoxysilylpropyl)tetrasulfide (BTESPT), while mesoporous T (tetrasulfide-based SiO_2_ NPs) were synthesized using the combination of TEOS and BTESPT. Nanoporous D (disulfide-based SiO_2_ NPs) was synthesized by using TEOS. (**b**). Doxorubicin (DOX) release from 110 Mesoporous D nanoparticles with and without GSH. Reprinted with permission from [40], ACS Publications, 2017.

**Figure 4 pharmaceutics-12-00890-f004:**
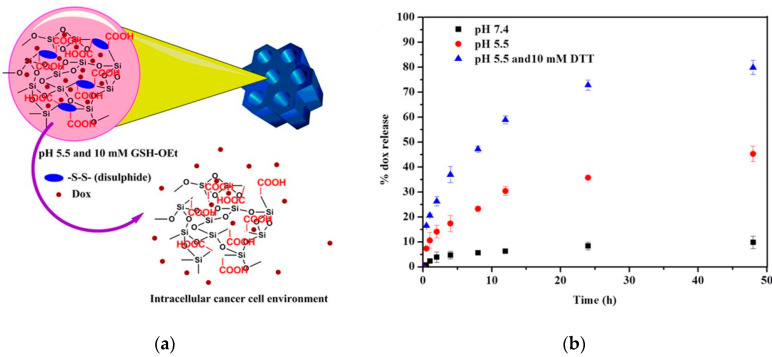
(**a**) Schematic representation of destabilization of cystamine-integrated periodic mesoporous organosilica (Cys-PMO) hybrid nanoparticles in response to redox and acidic pH environment and (**b**) in vitro DOX release profile from Cys-PMO hybrid nanoparticles. Reprinted with permission from [43], ACS Publications, 2017.

**Figure 5 pharmaceutics-12-00890-f005:**
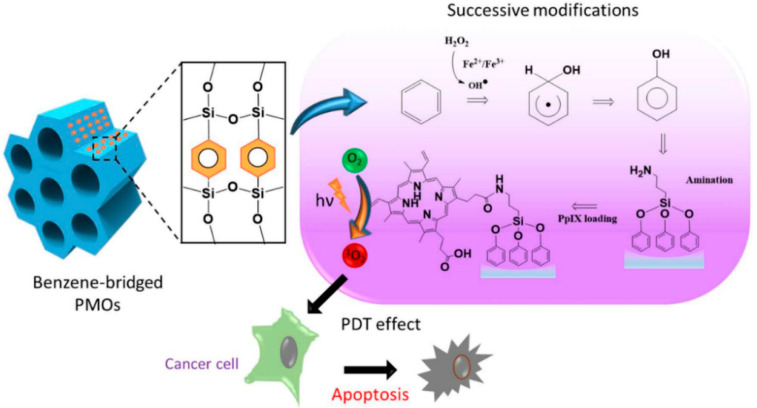
The reaction pathway of the hydroxylation of benzene and protoporphyrin IX loading and the mechanism elucidating the light-induced biological effects (Apoptosis). Reprinted from [56], MDPI Publications, 2020.

**Figure 6 pharmaceutics-12-00890-f006:**
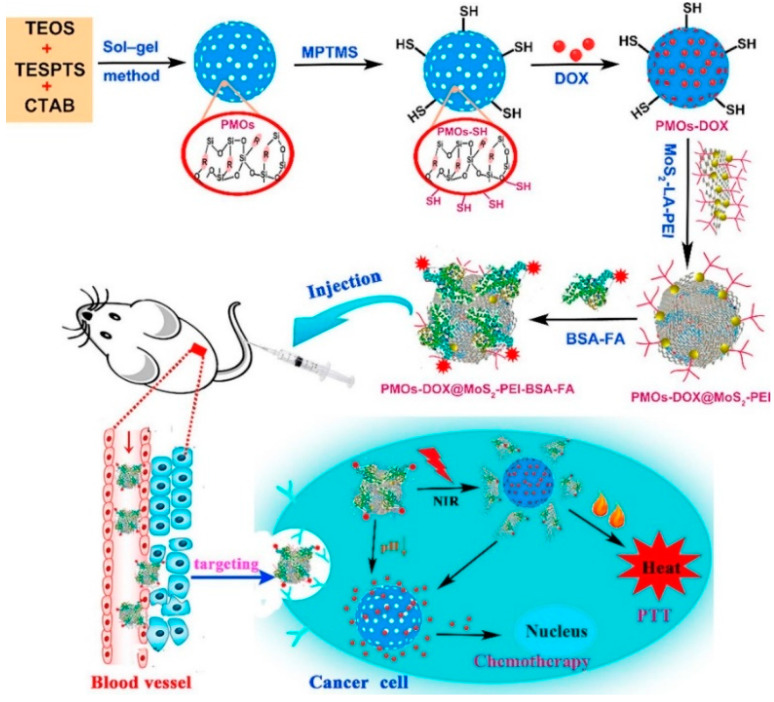
Schematic illustration for the synthesis and preparation of PMO–DOX@MoS_2_– lipoic acid (LA)–PEI–BSA–FA composite as a multifunctional drug delivery system for synergistic chemo-photothermal targeted therapy of tumors. Reprinted with permission from [57], Elsevier Publications, 2018.

**Table 1 pharmaceutics-12-00890-t001:** PMOs for drug delivery with internal and external stimuli.

PMOs	Organo Bridge	Drug	Loading Efficiency	Biodegradation/ Stimulus	Experiment	Reference
Nanorods and Nanosphere	Ethylene-bis(propyl)disulfide	Doxorubicin	22.20 wt%	pH	in vitro	[13]
Nanoparticles	ethylene–azidopropyl	Doxorubicin	14.20 wt%	pH and Two-photon	in vitro	[29]
Biodegradable PMO	bis [3-(triethoxysilyl) propyl] tetrasulfide	Doxorubicin	47.00 wt%	GSH	in vivo	[38]
Biodegradable PMO	bis [3-(triethoxysilyl) propyl] tetrasulfide	Daunorubicin	12.04 wt%	GSH	in vivo	[39]
CuS@PMOs-PEG/Nanosheets	Thioether	Doxorubicin	82.00 wt%	pH/Laser	in vivo	[41]
Cys PMO	Disulfide	Doxorubicin	50.60 wt%	pH, GSH	in vitro	[43]
PMO-NH_2_	1,2-bis(trimethoxysilyl)ethane	Doxorubicin	88.60 wt%	pH	in vitro	[45]
Yolk−shell PMOs	Thioether	Doxorubicin	31.70 wt%	GSH	in vivo	[50]
CuS@PMOs	1,4-Bis(triethoxysily)propane tetrasulfide	Doxorubicin	94.00 wt%	GSH	in vivo	[51]
DMPy-PMO	N,N’-Disilylated Pyridine	5-Fluorouracil	128 mg·g^−1^	pH	in vitro	[52]
Nanodiamond–PMO	Ethenylene and ethylene	Doxorubicin	40.00 wt%	Two-photon	in vitro	[55]
Nanoparticles	Benzene	Protoporphyrin IX	10.80 wt%	green laser light at 532 nm	in vitro	[56]
MoS_2_ nanosheet-capped PMOs	Thioether	Doxorubicin	185 mg·g^−1^	pH and NIR Light	in vivo	[57]
Hollow PMOs	Benzene	Doxorubicin	369 mg·g^−1^	ultrasound	in vivo	[58]

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
