# Peer review of "Recent Development to Explore the Use of Biodegradable Periodic Mesoporous Organosilica (BPMO) Nanomaterials for Cancer Therapy"

_pharmaceutics, 2020, doi:10.3390/pharmaceutics12090890_

Round 1
Reviewer 1 Report
In this review article, the authors summarized the recent development of biodegradable mesoporous silica nanomaterials for cancer therapy. The authors did a great job of collecting all related results in using silica nanoparticles for cancer treatment. This manuscript provides beneficial information for researchers working in this field. To help readers better understand these results, it will be a great idea to summarize the results in tables. We would like to suggest the authors add the following tables in the manuscript.
- The loading efficiency for different silica nanomaterials. The authors reported loading efficiency as part of material development. If the authors can list the loading efficiency for each material, the readers can compare the performance of the materials.
- The authors focused on the discussion of biodegradable silica nanomaterials. However, there are few discussions about the rate of degradation for different silica nanomaterials. If the authors can collect the degradation rate, the readers can choose the best candidate for their applications.
- Some experiments were conducted using cell lines, while others were performed in animal models. We would like to see a table showing the nanomaterials were tested in vitro, in vivo, or just pure material characterization.
Author Response
Comments
In this review article, the authors summarized the recent development of biodegradable mesoporous silica nanomaterials for cancer therapy. The authors did a great job of collecting all related results in using silica nanoparticles for cancer treatment. This manuscript provides beneficial information for researchers working in this field. To help readers better understand these results, it will be a great idea to summarize the results in tables. We would like to suggest the authors add the following tables in the manuscript.
Answer
Thank you for the comments. We have made changes as described below.
Comments
The loading efficiency for different silica nanomaterials. The authors reported loading efficiency as part of material development. If the authors can list the loading efficiency for each material, the readers can compare the performance of the materials.
Answer
We included loading efficiency for each material in Table 1.
Comments
The authors focused on the discussion of biodegradable silica nanomaterials. However, there are few discussions about the rate of degradation for different silica nanomaterials. If the authors can collect the degradation rate, the readers can choose the best candidate for their applications.
Answer
While the degradation rate of each material is important, this information is not available in the articles described. Therefore, we have not been able to include this information. Because the degradation rate varies on the conditions used, one needs to use a single condition with various materials and this has not been reported.
Comments
Some experiments were conducted using cell lines, while others were performed in animal models. We would like to see a table showing the nanomaterials were tested in vitro, in vivo, or just pure material characterization.
Answer
This information is included in Table 1.
Reviewer 2 Report
In my opinion Authors wrote a brief review that is well written and intersting.
However, a similar review has recently been published:
Trends in Degradable Mesoporous Organosilica-Based Nanomaterials for Controlling Drug Delivery: A Mini Review Materials (Basel) 020 Aug 19;13(17):E3668. doi: 10.3390/ma13173668.
Authors need to cite and discuss this review and demonstrate their novelty over it.
Limitations of EPR-based strategies should be discussed in the Introduction.
Degradation pictures/results should be added in Fig 3.
Authors should reproduce figures for pH, enyme and light responsive systems.
Authors need to write on biodegrability strategies and data of all the PMOs cited.
Author Response
Comments
In my opinion Authors wrote a brief review that is well written and interesting. However, a similar review has recently been published: Trends in Degradable Mesoporous Organosilica-Based Nanomaterials for Controlling Drug Delivery: A Mini Review Materials (Basel) 020 Aug 19;13(17): E3668. doi: 10.3390/ma13173668. Authors need to cite and discuss this review and demonstrate their novelty over it.
Answer
Thank you for mentioning the review by Poscher and Salinas. This review was published on August 19, 2020 during the time when our review was in preparation. There are differences between this and our review with respect to the topics covered (our review is focused more on mesoporous materials while this review discusses more non-porous materials). We mentioned this review in section 7.
Comments
Limitations of EPR-based strategies should be discussed in the Introduction.
Answer
We modified a sentence in the Introduction to mention that the tumor that can take advantage of the EPR mechanism needs to have rich vasculature.
Comments
Degradation pictures/results should be added in Fig 3.
Answer
A figure describing the degradation and release of the cargo is added in Figure 3.
Comments
Authors should reproduce figures for pH, enzyme and light responsive systems.
Answer
We added a new figure (Figure 4 In the revised version) that describes a pH/redox responsive system. A figure for a light responsive system is now figure 5
Comments
Authors need to write on biodegrability strategies and data of all the PMOs cited.
Answer
We added sentences to emphasize degradation strategies.